# Phenotypic and lifestyle determinants of HbA$_{1c}$ in the general population – The Hoorn Study

Willem Wisgerhof[1]*, Carolien Ruijgrok[1], Nicole R. den Braver[1], Karin J. Borgonjen—van den Berg[2], Amber A. W. A. van der Heijden[3], Petra J. M. Elders[3], Joline W. J. Beulens[1,4], Marjan Alssema[1,5]

1 Department of Epidemiology and Biostatistics, VU University Medical Center, Amsterdam Public Health Research Institute, Amsterdam, the Netherlands, 2 Department Agrotechnology and Food Sciences, Division of Human Nutrition, Wageningen University, Wageningen, the Netherlands, 3 Department of General Practice and Elderly Care Medicine, Amsterdam University Medical Centers, Amsterdam Public Health Research Institute, Amsterdam, the Netherlands, 4 Julius Center for Health Sciences and Primary Care, University Medical Center Utrecht, Utrecht, the Netherlands, 5 Health Council of the Netherlands, The Hague, the Netherlands

☯ These authors contributed equally to this work.
* willemwisgerhof@gmail.com

**Data Availability Statement:** Data cannot be shared publicly because of sensitive patient information. Data are available from the Ethics Committee of the VU University Medical Center

## Abstract

### Aim

To investigate the relative contribution of phenotypic and lifestyle factors to HbA$_{1c}$, independent of fasting plasma glucose (FPG) and 2h post-load glucose (2hPG), in the general population.

### Methods

The study populations included 2309 participants without known diabetes from the first wave of the Hoorn Study (1989) and 2619 from the second wave (2006). Multivariate linear regression models were used to analyze the relationship between potential determinants and HbA$_{1c}$ in addition to FPG and 2hPG. The multivariate model was derived in the first wave of the Hoorn Study, and replicated in the second wave.

### Results

In both cohorts, independent of FPG and 2hPG, higher age, female sex, larger waist circumference, and smoking were associated with a higher HbA$_{1c}$ level. Larger hip circumference, higher BMI, higher alcohol consumption and vitamin C intake were associated with a lower HbA$_{1c}$ level. FPG and 2hPG together explained 41.0% (first wave) and 53.0% (second wave) of the total variance in HbA$_{1c}$. The combination of phenotypic and lifestyle determinants additionally explained 5.7% (first wave) and 3.9% (second wave).

(contact via metc@vumc.nl) for researchers who meet the criteria for access to confidential data.

**Funding:** This study was financially supported by a grant from Unilever R&D Vlaardingen, the Netherlands (https://www.unilever.com/) and a grant from the Amsterdam Public Health Research Institute, Amsterdam, the Netherlands (https://www.amsterdamresearch.org/web/public-health/home.htm). Unilever also provided support in the form of a salary for MA, who was an employee at the time this article was written. The specific role of this author is articulated in the 'author contributions' section. The funders had no role in study design, data collection and analysis, decision to publish, or preparation of the manuscript. https://www.amsterdamresearch.org/web/public-health/home.htm The funders had no role in study design, data collection and analysis, decision to publish, or preparation of the manuscript.

**Competing interests:** The authors have read the journal's policy and have the following competing interests: At the time this article was written, MA was a paid employee of Unilever, which manufactures and sells consumer food products. This does not alter our adherence to PLOS ONE policies on sharing data and materials. There are no patents, products in development or marketed products associated with this research to declare. All other authors declare that they have no potential conflict of interests relevant to this article.

## Conclusions

This study suggests that, independent of glucose, phenotypic and lifestyle factors are associated with HbA$_{1c}$, but the contribution is relatively small. These findings contribute to a better understanding of the low correlation between glucose levels and HbA$_{1c}$ in the general population.

## Introduction

Glycated haemoglobin (HbA$_{1c}$) is widely adopted as diagnostic marker for type 2 diabetes and reflects average plasma glucose levels over the prior two to three months [1, 2]. The use of HbA$_{1c}$ for diagnosis has several advantages over measurements of fasting plasma glucose (FPG) and 2h post-load plasma glucose (2hPG), which include lower intra-individual variation and more convenience in blood sampling conditions [3]. HbA$_{1c}$ has also been shown to be a more accurate marker than measures of glucose for future risk of diabetes complications, such as micro vascular complications and cardiovascular disease (CVD), in individuals with type 2 diabetes [4]. In addition, higher HbA$_{1c}$ is associated with higher risk for CVD and total mortality [5, 6]. This increased CVD and mortality risk with higher HbA$_{1c}$ is already seen at levels in the normal range [5, 6].

Nonetheless, controversy exists regarding the use of HbA$_{1c}$ as a diagnostic marker for type 2 diabetes in the general population, since several population-based studies have shown that correlations between glucose and HbA$_{1c}$ are relatively poor [7]. Furthermore, several studies have suggested that HbA$_{1c}$ levels are affected by other factors than glucose alone [8–10]. Multiple studies found that HbA$_{1c}$ might not be a reliable reflection of glycemic state in individuals with iron or vitamin B12 deficiency, renal or liver failure, short erythrocyte lifespan, rheumatoid arthritis, alcoholism, and in individuals who use aspirin, vitamin -C, -E supplements or antiretrovirals [11, 12]. To better understand the effects of such conditions on HbA$_{1c}$, a more comprehensive understanding of determinants of HbA$_{1c}$ is needed.

A small number of studies revealed that, independent of glucose, non-Hispanic black race, smoking, several genetic risk factors and higher age and BMI are associated with a higher HbA$_{1c}$ level, whereas higher alcohol consumption, haemoglobin mass and concentration are associated with a lower HbA$_{1c}$ level [13–16]. Another previous study showed that in individuals without diabetes, 2hPG explains only one quarter to one third of the variance in HbA$_{1c}$ [9]. Furthermore, researchers of the Lifeline Cohort Study recently found that a combination of clinical, lifestyle and genetic factors, in addition to FPG, explained 26.2% of the variance in HbA$_{1c}$, with FPG contributing 10.9% [13]. Exploring determinants in addition to both FPG and 2hPG provides better insight into the added value of other determinants, since the combination of FPG and 2hPG is a better reflection of glycemic state and diurnal glucose [17].

A better understanding of the variance in HbA$_{1c}$ is relevant for the interpretation of HbA$_{1c}$ levels in the general population. The present study therefore aims to investigate the relative contribution of previously identified determinants to HbA$_{1c}$, independent of FPG and 2hPG, in the general population.

## Subjects, materials and methods

### Study population

The current study was performed using baseline data from the Hoorn Study, a population based cohort study conducted among men and women from the general population residing

in the city of Hoorn, the Netherlands [18]. For the first wave of the Hoorn Study a total of 3553 participants aged between 49 and 75 years were invited and 2540 agreed to participate in the study, of which 56 non-Caucasians were excluded. In the present study, we excluded 90 participants with known diabetes based on GP diagnosis and/or use of glucose-lowering medication at baseline and 15 participants with missing data on diabetes status. Furthermore, we excluded 69 participants without data on dietary intake variables and one participant with an extreme value for total energy (<500 or >5000 kcal/day), leaving 2309 participants for analysis. Baseline data were collected from October 1989 until February 1992.

The second wave was conducted July 2006 until November 2007 and a total of 6180 men and women aged 40–65 years were invited, of which 2807 agreed to participate in the study [18]. In the present study, we excluded 82 participants with known diabetes based on GP diagnosis and/or use of glucose-lowering medication at baseline and 54 participants with missing data on diabetes status. Furthermore, we excluded 43 participants without data on dietary intake variables and nine participants with extreme values for total energy (<500 or >5000 kcal/day), leaving 2619 participants for analysis. Informed consent was signed by all participants and both studies were approved by the Ethical Review Committee of the VU University medical center (VUmc).

## Baseline measurements

In the first wave, anthropometric measurements were taken at the first visit with participants being barefoot and wearing light clothes [18]. Data on dietary intake and alcohol consumption were obtained from a 75-item self-reported semi quantitative food frequency questionnaire (FFQ) [19]. Energy and nutrient intakes were estimated from FFQ through linkage with a computerized version of the Dutch Food Composition Table [20]. The FFQ was validated against a dietary history for total energy intake, all macronutrients, dietary fiber, alcohol, iron, vitamins B1, B6 and vitamin C. Pearson correlation coefficients were on average 0.71 (range: 0.65–0.78) for macronutrients, and 0.66 (range: 0.36–0.81) for vitamins and minerals [19]. Cigarette smoking status was self-reported.

In the second wave, the same methods as in the first wave were used for anthropometric measurements and cigarette smoking status [18]. Data on dietary intake and alcohol consumption were obtained from a 125-item self-reported quantitative FFQ. The Dutch Food Composition Table of 2006 was used to calculate energy and nutrient intake per day [21]. This FFQ was validated for total energy intake against actual energy intake needed to maintain stable body weights during 11 controlled dietary trials, with a Pearson correlation coefficient of 0.82 (95% CI, 0.80, 0.85) [22].

## Laboratory assays

In the first wave, an oral glucose tolerance test (OGTT) was performed after an overnight fast and blood samples were collected before and 2 hours after intake of the glucose load [18]. FPG and 2hPG levels were determined with the glucose dehydrogenase method (Merck, Darmstadt, Germany). HbA$_{1c}$ was assessed with an ion exchange high performance liquid chromatography (Modular Diabetes Monitoring System, BioRad Lab, Veenendaal, The Netherlands: normal range 4.3–6.1%). Alanine transaminase enzyme activity was measured in plasma as a marker of liver function [23]. This measurement was conducted according to the method of the International Federation of Clinical Chemistry from 1985 and expressed as IU/L. Serum creatinine was measured as indicator of renal function [24].

In the second wave, the same methods were used for the OGTT [18]. FPG and 2hPG levels were measured with the glucose oxidase method (Boehringer-Mannheim, Mannheim,

Germany). HbA$_{1c}$ was assessed using the Diabetes Control and Complications Trial (DCCT) standardized reverse-phase cation exchange chromatography (Menarini, Florence, Italy), for which the intra-assay coefficient of variation was 0.65% at a mean of 4.89%, and the interassay coefficient of variation was 1.55% at a mean of 5.52% [25]. In both studies, all analyses were performed at the clinical chemistry laboratory of the VUmc.

## Statistical analysis

Characteristics of the study populations are presented as mean ± SD for normally distributed continuous variables or median (25$^{th}$– 75$^{th}$ percentile) for positively skewed distributions. Categorical variables are presented as proportions. Missing data (0.1% of all observations in the first wave and 0.2% in the second wave) were imputed using multiple imputation, creating five imputed datasets. The Multivariate Imputation by Chained Equations (MICE) algorithm was used with the predictive mean matching method [26]. Results of the analyses were pooled using Rubin's rules [27].

Correlations between FPG, 2hPG and HbA$_{1c}$ were determined using Spearman correlations. Linear and quadratic associations of the determinants with HbA$_{1c}$ were added in the linear regression model to test for non-linear associations. Determinants for which the quadratic term was significant ($P < 0.05$) were divided into quartiles. Dietary intake variables were corrected for total energy intake by the residual method [28].

Using data from the first wave, a multivariate linear regression model with outcome HbA$_{1c}$ was created including FPG and 2hPG as fixed determinants, and age, sex, BMI, waist and hip circumference, heamaglobin, serum creatinine, serum alanine transaminase, smoking, alcohol consumption, carbohydrate intake, fiber intake, iron intake, vitamin b1 and b6 intake, vitamin C intake as potential determinants. Determinants in the multivariate model were included based on a stepwise backward selection procedure with $P < 0.10$ considered statistically significant. Continuous determinants were standardized into a Z-score. The set of significant predictors in the multivariate model constructed within the first wave was replicated as a multivariate model in the data of the second wave. Variables were considered as determinants of HbA$_{1c}$ if there was a significant association in the multivariable model in both cohorts. Effect-modification was investigated by including interaction terms between sex and other independent variables in the linear regression models. Interaction terms were considered statistically significant with $P < 0.10$. The explained variance of the models was estimated by the median R$^2$ of the imputed datasets [29]. All analyses were performed using IBM SPSS Statistics version 22 for Windows (SPSS, Chicago, IL, USA).

## Results

Table 1 presents the population characteristics of the first (1989) and the second (2006) wave of the Hoorn Study. At the time of examination, participants from the first wave were on average 7.5 years older (61.4 ± 7.3) than those of the second wave (53.9 ± 6.7). Furthermore, the proportion of current smokers was substantially greater in the first (31.5%) than in the second wave (21.0%), whereas the median alcohol consumption in the second wave (8.2 g/day) was approximately two times higher than in the first wave (4.3 g/day). The Spearman correlations (Table 1) of FPG with HbA$_{1c}$ were stronger in both waves of the Hoorn Study (first: $r = 0.32$; second: $r = 0.42$) than the correlations of 2hPG with HbA$_{1c}$ (first: $r = 0.22$; second: $r = 0.31$). Of all participants who were diagnosed with diabetes according to at least one of the screening tools (FPG $\geq$ 7.0 mmol/l or 2hPG $\geq$ 11.1 mmol/l or HbA$_{1c}$ $\geq$ 6.5% (48 mmol/mol)), 23.8% (first) and 12.1% (second) were diagnosed with diabetes according to all three screening tools (Fig 1).

**Table 1. Population characteristics and Spearman correlations of the participants of the Hoorn Study.**

| | First wave | Second wave |
|---|---|---|
| N | 2309 | 2619 |
| Age (years) | 61.4 ± 7.3 | 53.9 ± 6.7 |
| Female sex (%) | 54.2 | 54.0 |
| BMI (kg/m$^2$) | 26.5 ± 3.5 | 26.1 ± 3.9 |
| Waist circumference (cm) | 90.6 ± 10.7 | 89.4 ± 11.4 |
| Hip circumference (cm) | 101.6 ± 6.7 | 100.3 ± 7.9 |
| Fasting plasma glucose (mmol/l) | 5.6 ± 1.0 | 5.5 ± 0.8 |
| 2h post-load glucose (mmol/l) | 6.1 ± 2.8 | 5.7 ± 2.0 |
| HbA$_{1c}$ (mmol/mol) | 36.0 ± 7.7 | 36.0 ± 4.4 |
| HbA$_{1c}$ (%) | 5.4 ± 0.7 | 5.4 ± 0.4 |
| Haemoglobin (mmol/l) | 9.0 ± 0.8 | N/A |
| Serum alanine transaminase (IU/l) | 11.0 (8.0–15.0) | N/A |
| Serum creatinine (μmol/l) | 90.1 ± 15.6 | N/A |
| Current smoker (%) | 31.5 | 21.0 |
| Total energy intake (kcal/day) | 2067.6 ± 581.5 | 2128.4 ± 665.0 |
| Alcohol consumption (g/day) | 4.3 (0.0–12.5) | 8.2 (2.1–19.6) |
| Carbohydrate intake (g/day) | 213.9 ± 65.8 | 243.6 ± 80.5 |
| Fiber intake (g/day) | 27.2 ± 7.9 | N/A |
| Iron intake (mg/day) | 12.5 ± 3.1 | 11.0 ± 3.0 |
| Vitamin B1 intake (mg/day) | 1.1 ± 0.3 | 1.2 ± 0.4 |
| Vitamin B6 intake (mg/day) | 1.4 ± 0.4 | 1.7 ± 0.5 |
| Vitamin C intake (mg/day) | 95.3 (67.2–134.1) | 92.0 (67.1–119.7) |
| FPG vs. HbA$_{1c}$[a] | 0.32[b] | 0.42[b] |
| 2hPG vs. HbA$_{1c}$[a] | 0.22[b] | 0.31[b] |
| FPG vs. 2hPG[a] | 0.42[b] | 0.37[b] |

N/A, not available; Values are means ± SD, median (25$^{th}$–75$^{th}$ percentile) or percentage.

[a]Spearman correlations.

[b]Statistically significant ($P < 0.05$) correlation.

Table 2 shows the results of the multivariate linear regression models for both the first and the second wave (replication analysis). No interactions ($P > 0.10$ for interaction terms) were found between sex and other independent variables in the multivariate model. In the first wave, independent of glucose, significant associations were found for age, sex, BMI, waist circumference, hip circumference, serum creatinine, smoking, alcohol consumption, carbohydrate intake, fiber intake and vitamin C intake in the multivariate model with HbA$_{1c}$. Of these variables, age, sex, BMI, waist circumference, hip circumference, smoking, alcohol consumption and vitamin C intake were confirmed as statistically significant associations in the second wave. In the second wave, one SD increase in age (6.7 years) was associated with a 0.03% (95% CI, 0.03, 0.04) higher HbA$_{1c}$ level. On average females had a 0.08% (0.07, 0.09) higher HbA$_{1c}$ level than men. The quartiles of BMI and hip circumference showed a negative association with HbA$_{1c}$, while waist circumference was positively associated. Smokers had a 0.10% (0.09, 0.11) higher HbA$_{1c}$ level than non-smokers, whereas alcohol consumption was associated with a 0.07% (-0.07, -0.06) lower HbA$_{1c}$ level per one SD increase (15.1 g/day). Quartiles of vitamin C intake were inversely associated with HbA$_{1c}$. The multivariate association found in the first wave for carbohydrate intake was not confirmed in the second wave. Data on creatinine and fiber intake were not available in the second wave.

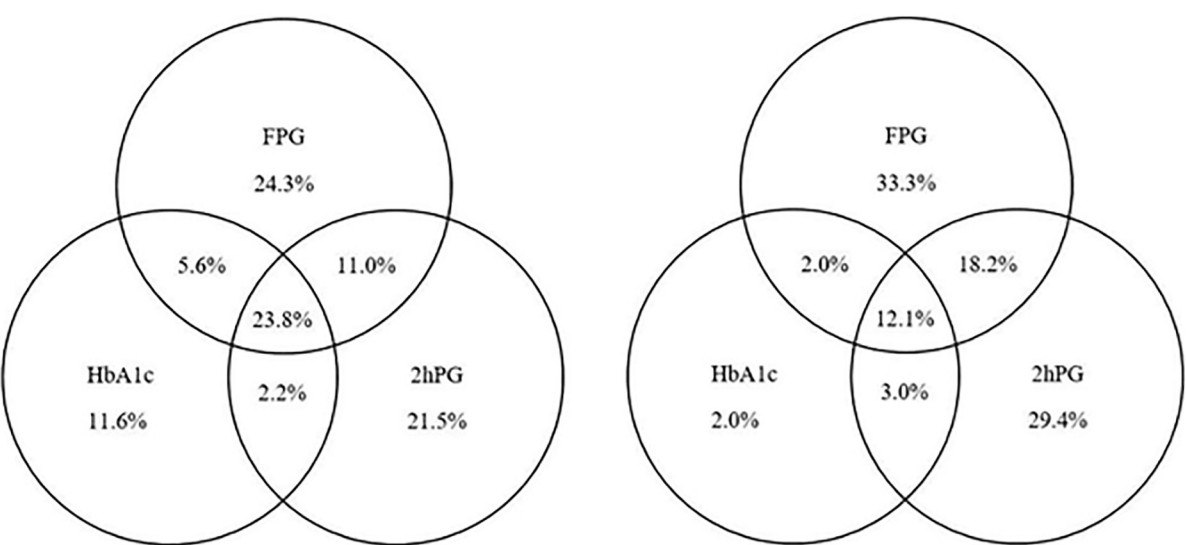

**Fig 1. Diagnosis of diabetes.** The percentage of participants diagnosed with diabetes in the first wave *(n = 181 out of 2309)* and the second *wave (n = 108 out of 2612)*. Diabetes mellitus was diagnosed when at least one of the following criteria were met: FPG ≥ 7.0 mmol/l; 2hPG ≥ 11.1 mmol/l; HbA$_{1c}$ ≥ 48 mmol/mol (6.5%).

FPG alone explained 39.5% (first) and 51.2% (second) of the total variance in HbA$_{1c}$. FPG and 2hPG together explained 41.0% and 53.0% of variance in HbA$_{1c}$. Phenotypic and lifestyle determinants in combination with both FPG and 2hPG explained an additional 5.7% in the first wave and 3.9% in the second wave. Models containing phenotypic and lifestyle determinants without glucose explained 11.1% and 10.3% of variance in HbA$_{1c}$.

## Discussion

The current study investigated potential determinants of HbA$_{1c}$, independent of FPG and 2hPG. In two independent cohorts, we found that higher age, female sex, larger waist circumference and smoking were associated with a higher HbA$_{1c}$ level, whereas larger hip circumference, higher BMI, higher alcohol consumption and vitamin C intake were associated with a lower HbA$_{1c}$ level. The key finding of this study was that glycemic variables explained the vast majority of the variation in HbA$_{1c}$, while the contribution of phenotypic and lifestyle determinants was relatively small.

Our findings that age, sex, BMI, waist and hip circumference, smoking, alcohol consumption and vitamin C intake are associated with HbA$_{1c}$ in two independent cohorts is consistent with available evidence. In line with previous studies [13, 30, 31], we observed that age and smoking are associated with HbA$_{1c}$ independent of glucose. The current study reported a higher HbA$_{1c}$ level in women than in men, after correction of glucose levels. Previous studies that did not account for glucose reported higher levels of HbA$_{1c}$ in men, which is assumed to be the result of a longer erythrocyte lifespan in men [16, 32]. Similarly, we observed a higher HbA$_{1c}$ level in men when we did not correct for glucose. In the present study we found independent and opposite associations of waist and hip circumference with HbA$_{1c}$. Larger waist, and smaller hip circumference has previously been shown to be associated with risk for developing type 2 diabetes, independent of BMI [33, 34]. Our current findings showing associations between body composition and HbA$_{1c}$, independent of glucose measures are novel. The same is true for alcohol consumption which has been associated with a lower HbA$_{1c}$ level and improved insulin resistance in previous observational and intervention studies, but not

**Table 2. Multivariate linear regression models of associations of demographic, anthropometric, clinical and nutritional determinants with HbA1c independent of glucose in the Hoorn Study.**

| | | First wave (*n* = 2309) | | | Second wave (*n* = 2619) | | |
| --- | --- | --- | --- | --- | --- | --- | --- |
| | | HbA$_{1c}$ (% of Hb) | | | HbA$_{1c}$ (% of Hb) | | |
| | | Beta | 95% CI | *P*-value | Beta | 95% CI | *P*-value |
| FPG (mmol/l) per SD = 1.0 in first wave / 0.8 in second wave | | 0.353 | 0.299;0.358 | <0.001 | 0.272 | 0.266;0.278 | <0.001 |
| 2hPG (mmol/l) per SD = 2.9 / 2.0 | | 0.109 | 0.079;0.139 | <0.001 | 0.060 | 0.054;0.065 | <0.001 |
| Age per SD = 7.3 / 6.7 | | 0.052 | 0.031;0.073 | <0.001 | 0.031 | 0.027;0.036 | <0.001 |
| Sex F/M (ref) | | 0.096 | 0.036;0.155 | 0.002 | 0.081 | 0.069;0.092 | <0.001 |
| BMI (kg/m$^2$) | Q1 | 0 | | | 0 | | |
| | Q2 | -0.119 | -0.182;-0.057 | <0.001 | -0.015 | -0.029;-0.001 | 0.039 |
| | Q3 | -0.094 | -0.165;-0.022 | 0.011 | -0.016 | -0.033;0.001 | 0.068 |
| | Q4 | -0.135 | -0.228;-0.043 | 0.004 | -0.022 | -0.044;0.001 | 0.061 |
| Waist circumference (cm) per SD = 10.7 / 11.4 | | 0.047 | 0.012;0.081 | 0.009 | 0.015 | 0.006;0.024 | 0.001 |
| Hip circumference (cm) | Q1 | 0 | | | 0 | | |
| | Q2 | -0.058 | -0.117;0.002 | 0.059 | -0.013 | -0.027;0 | 0.057 |
| | Q3 | -0.002 | -0.068;0.065 | 0.953 | -0.029 | -0.045;-0.013 | <0.001 |
| | Q4 | -0.025 | -0.105;0.055 | 0.537 | -0.015 | -0.035;0.005 | 0.131 |
| Serum creatinine (μmol/l) | Q1 | 0 | | | N/A | | |
| | Q2 | 0.023 | -0.033;0.080 | 0.417 | N/A | | |
| | Q3 | 0.055 | -0.007;0.116 | 0.081 | N/A | | |
| | Q4 | 0.092 | 0.024;0.160 | 0.008 | N/A | | |
| Current smoking Y/N (ref) | | 0.236 | 0.191;0.281 | <0.001 | 0.101 | 0.090;0.112 | <0.001 |
| Alcohol consumption[a] per SD = 12.4 / 15.1 | | -0.086 | -0.108;-0.063 | <0.001 | -0.065 | -0.070;-0.059 | <0.001 |
| Carbohydrate intake[a] per SD = 65.8 / 80.5 | | -0.028 | -0.050;-0.005 | 0.018 | 0.000 | -0.005;0.005 | 0.996 |
| Fiber intake[a] per SD = 7.9 / NA | | 0.020 | -0.004;0.044 | 0.096 | N/A | | |
| Vitamin C intake[a] | Q1 | 0 | | | 0 | | |
| | Q2 | -0.057 | -0.114;-0.001 | 0.048 | -0.032 | -0.045;-0.020 | <0.001 |
| | Q3 | -0.075 | -0.134;-0.016 | 0.013 | -0.040 | -0.053;-0.027 | <0.001 |
| | Q4 | -0.113 | -0.175;-0.051 | <0.001 | -0.038 | -0.052;-0.025 | <0.001 |

N/A, not available; Q, Quartile (range); First wave; BMI, Q1 (<24.1) Q2 (24.1;26.1) Q3 (26.2;28.3) Q4 (>28.3); Hip circumference, Q1 (<97.5) Q2 (97.5;101.9) Q3 (102.0;106.7) Q4 (>106.7) Serum creatinine, Q1 (<79.0) Q2 (79.0;88.0) Q3 (88.1;98.0) Q4 (>98.0); Vitamin C, Q1 (< -0.7) Q2 (-0.7;-0.2) Q3 (-0.1;0.6) Q4 (>0.6).
Second wave; BMI, Q1(<23.4) Q2 (23.4;25.6) Q3(25.7;28.2) Q4 (>28.2); Hip circumference, Q1 (<95.4) Q2 (95.4;99.8) Q3 (99.9;105.2) Q4 (>105.2); Vitamin C, Q1 (< -0.7) Q2 (-0.7; -0.1) Q3 (0.0; 0.6) Q4 (>0.6); Multivariate linear regression models regarding potential determinants and outcome HbA$_{1c}$, in addition to fasting plasma glucose and 2h post-load glucose.
First wave complete model; R$^2$ 46.7%. FPG + 2hPG; R$^2$ 41.0%. (non-glycemic additional 5.7%) FPG; R$^2$ 39.5%. 2hPG; R$^2$ 29.7%. Phenotypic and lifestyle determinants; R$^2$ 11.1%. Second wave complete model; R$^2$ 56.9%. FPG + 2hPG; R$^2$ 53.0%. (non-glycemic additional 3.9%) FPG; R$^2$ 51.2%. 2hPG; R$^2$ 27.9%. Phenotypic and lifestyle; R$^2$ 10.3%.
[a]Corrected for total energy intake by the residual method.

particularly with HbA$_{1c}$ independent of glucose [35–37]. The mechanism by which alcohol would improve insulin sensitivity is currently not resolved. Similarly, our finding that vitamin C intake was associated with a lower HbA$_{1c}$ level has been reported in previous observational studies, but not independent of glucose [38, 39]. As suggested by Davie et al. [40], this relationship is possibly not mediated by glucose but caused by a competition between vitamin C and glucose to react with the protein amino group, thereby reducing glycation of haemoglobin.

In both the first (46.7%) and second wave (56.9%) model we found a substantially higher explained variance in HbA$_{1c}$ as compared to a multivariate model created by Jansen et al. (26.2%), which contained some phenotypic, clinical, lifestyle and genetic determinants, in

addition to FPG [13]. This difference is already apparent in a model with FPG alone, which explained 10.9% of the variance in HbA$_{1c}$ in the study by Jansen et al., whereas it explained 39.5% (first) and 51.2% (second) in the current study. One reason for this difference may be the larger variation in HbA$_{1c}$ in our study compared to Jansen et al. The standard deviation of HbA$_{1c}$ was 0.7% and 0.4% in the Hoorn Study, while this was only 0.3% in the study by Jansen et al.

The present study and previous studies indicate that HbA$_{1c}$ may reflect in part aging processes, body composition, smoking and dietary habits. These findings contribute to better understanding of the relatively poor correlations between glucose and HbA$_{1c}$ observed in a previous investigation in the second wave [25] and other population-based studies [7]. In individuals with levels of HbA$_{1c}$ that are higher or lower than expected based on glucose levels, the present data may provide clues for explaining such disagreement. Furthermore, presently identified determinants of HbA$_{1c}$ might explain the stronger relationship with CVD observed in previous studies [4–6]. Future prospective studies are needed to better understand which factors explain or mediate the relationship between HbA1c and CVD.

The main strength of our study is the investigation of a combined set of potential determinants of HbA$_{1c}$, in addition to both FPG and 2hPG, which provides a better reflection of diurnal glucose than FPG alone. Another strength of this study is the selection of potential determinants identified by previous studies, and the replication of the analyses in independent data, both limiting the probability of chance-findings. Our study also has some limitations. Firstly, several potential determinants of HbA$_{1c}$, such as erythrocyte lifespan, vitamin B12 and E, genetic factors, rheumatoid arthritis and antiretrovirals could not be taken into account. Particularly erythrocyte lifespan could be of significant influence on HbA$_{1c}$ levels because the glycation of haemoglobin partially depends on it [11]. Secondly, data on serum creatinine and fiber intake were not available in the second wave. Therefore, the model of the first wave could not be exactly replicated in the second wave.

In summary, the current study shows that, in addition to both FPG and 2hPG, age, sex, BMI, waist and hip circumference, smoking, alcohol consumption and vitamin C intake are associated with HbA$_{1c}$ in the general population. Although the variation in HbA$_{1c}$ explained by these factors is relatively low, it suggests that variation in HbA$_{1c}$ levels apart from glucose may also be partially determined by phenotypic and lifestyle factors. These findings contribute to better understanding of the low correlation between glucose levels and HbA$_{1c}$ in the general population.

## Acknowledgments

We would like to thank WUR for dietary intake data linkage with NEVO and calculations.

## Author Contributions

**Conceptualization:** Willem Wisgerhof, Carolien Ruijgrok, Petra J. M. Elders, Joline W. J. Beulens, Marjan Alssema.

**Data curation:** Nicole R. den Braver, Karin J. Borgonjen—van den Berg, Amber A. W. A. van der Heijden, Petra J. M. Elders, Joline W. J. Beulens.

**Formal analysis:** Willem Wisgerhof, Carolien Ruijgrok.

**Funding acquisition:** Joline W. J. Beulens.

**Investigation:** Carolien Ruijgrok, Petra J. M. Elders, Joline W. J. Beulens.

**Methodology:** Willem Wisgerhof, Carolien Ruijgrok, Joline W. J. Beulens, Marjan Alssema.

**Project administration:** Nicole R. den Braver, Karin J. Borgonjen—van den Berg, Amber A. W. A. van der Heijden, Petra J. M. Elders, Joline W. J. Beulens.

**Resources:** Nicole R. den Braver, Karin J. Borgonjen—van den Berg, Amber A. W. A. van der Heijden, Petra J. M. Elders, Joline W. J. Beulens.

**Software:** Willem Wisgerhof, Carolien Ruijgrok.

**Supervision:** Joline W. J. Beulens, Marjan Alssema.

**Validation:** Marjan Alssema.

**Visualization:** Willem Wisgerhof, Carolien Ruijgrok.

**Writing – original draft:** Willem Wisgerhof, Carolien Ruijgrok, Marjan Alssema.

**Writing – review & editing:** Carolien Ruijgrok, Nicole R. den Braver, Karin J. Borgonjen—van den Berg, Amber A. W. A. van der Heijden, Petra J. M. Elders, Joline W. J. Beulens, Marjan Alssema.

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
