## [Decision Letter · Decision Letter 0]

25 Mar 2020

PONE-D-19-25704

Phenotypic and lifestyle determinants of HbA1c in the general population – the Hoorn Studies

PLOS ONE

Dear Dr Wisgerrhod

Thank you for submitting your manuscript to PLOS ONE. After careful consideration, we feel that it has merit but does not fully meet PLOS ONE’s publication criteria as it currently stands. Therefore, we invite you to submit a revised version of the manuscript that addresses the points raised during the review process.

We would appreciate receiving your revised manuscript by April 30. To enhance the reproducibility of your results, we recommend that if applicable you deposit your laboratory protocols in protocols.io, where a protocol can be assigned its own identifier (DOI) such that it can be cited independently in the future. For instructions see: http://journals.plos.org/plosone/s/submission-guidelines#loc-laboratory-protocols

We look forward to receiving your revised manuscript.

Kind regards,

Xianwu Cheng, M.D., Ph.D., FAHA

Academic Editor

PLOS ONE

Journal Requirements:

"I have read the journal's policy and the authors of this manuscript have the following

competing interests: at the time this article was written, Marjan Alssema was employed

by Unilever which manufactures and sells consumer food products. All other authors

declare that they have no potential conflict of interests relevant to this article."

Additional Editor Comments (if provided):

None.

Reviewers' comments:

Reviewer's Responses to Questions

**Comments to the Author**

1. Is the manuscript technically sound, and do the data support the conclusions?

Reviewer #1: Yes

Reviewer #2: Yes

2. Has the statistical analysis been performed appropriately and rigorously? 

Reviewer #1: Yes

Reviewer #2: Yes

3. Have the authors made all data underlying the findings in their manuscript fully available?

Reviewer #1: No

Reviewer #2: Yes

4. Is the manuscript presented in an intelligible fashion and written in standard English?

Reviewer #1: Yes

Reviewer #2: Yes

5. Review Comments to the Author

Reviewer #1: The manuscript is technically reasonable, the data is reliable, and appropriate and rigorous statistical analysis is performed, but the conclusions need to be further proved. The manuscript is presented in an intelligible fashion and written in standard English, but the author failed to provide important factors such as hemoglobin and serum alanine aminotransferase in the manuscript. Needs further research and reveals.

Reviewer #2: General Comments

This is a very important and interesting study. The replication element of the study greatly increases its impact. It is important to determine the relation between the potential determinants of HbA1c and HbA1c itself.

Specific comments

Methods

A breakdown of Ethnicity of the participants would be very helpful – sorry if I missed this

Results

Please can a Figure be included to display the results of the multivariate regression analysis as described in Table 2. This always aids data appreciation from a reader perspective.

6. PLOS authors have the option to publish the peer review history of their article (what does this mean?). If published, this will include your full peer review and any attached files.

Reviewer #1: No

Reviewer #2: Yes: Dr. Adrian Heald

---

## [Author Response · Author response to Decision Letter 0]

24 Apr 2020

Reviewer #1:

Summary: 

This manuscript by Willem et al. entitled “Phenotypic and lifestyle determinants of HbA1c in the general population – the Hoorn Studies” is aimed at understanding the relevance of phenotypic and lifestyle factors to HbA1c, in the general population. The data of both cohorts, independent of FPG and 2hPG, higher age, female sex, larger waist circumference, and smoking were associated with a higher HbA1c level. Larger hip circumference, higher BMI, higher alcohol consumption and vitamin C intake were associated with a lower HbA1c level. FPG and 2hPG together explained 41.0% (HS) and 53.0% (NHS) of the total variance in HbA1c. The authors fully describe the relationship between lifestyle, blood glucose levels and glycated hemoglobin in NHS and HS. There are several concerns to consider.

Major:

This study is simply a description and fails to give a reasonable explanation. The results of BMI related to increased hip circumference and low-grade glycated hemoglobin are interesting, but there is no explanation. In general, glycated hemoglobin is a non-independent diagnostic standard for diabetes. In this study, as a key point to explore, the actual or predictive role of glycated hemoglobin changes in the model population should be more revealed. At present, I cannot appreciate the value of the information provided by the research to clinical or public nutrition research. I hope researchers can further discuss and reveal the significance of the research. 

Response: We thank the reviewer for careful reading of the paper and for addressing these important points. We will respond to these in a point-by-point answer below. 

1．The trend of BMI results in HS from Q1 to Q4 is inconsistent. Can this result be adopted?

Response: the trend over quartiles BMI in the HS cohort is indeed slightly inconsistent. This inverse association of BMI with HbA1c is partly due to the model being multivariate and including hip and waist circumference as well as fasting and post-load glucose. This will make closely related estimates somewhat unstable. Therefore, we avoided to speculate on the importance of this slightly inconsistent association over BMI quartiles.

2. Hemoglobin and serum alanine aminotransferase are important factors affecting glycated hemoglobin, which are not shown in the analysis results, can they be explained?

Response: Hemoglobin and serum alanine aminotransferase were indeed considered potential determinants of HbA1c in addition to glucose measures. These factors were added to the multivariable model of the HS cohort in a stepwise procedure. However both did not appear statistical significant determinants at level P< 0.10 and were therefore not presented in Table 2. Serum alanine transferase indeed was an important determinant, that was excluded in the final model with a p-value of 0.25, while hemoglobin was already excluded earlier in the analysis. 

3. Can I know the results about the investigator's medication history?

Response: the medication history for blood glucose lowering medication was considered in the study. All study participants with diabetes and thus also those who were using blood glucose lowering medication were excluded from the analysis. This is indicated more clearly now on page 5 line 98 and 106. 

4. The author's study showed a negative correlation between the amount of alcohol consumed and the content of glycated hemoglobin, which means that increased drinking in healthy people is beneficial to prevent blood sugar rising. Can I understand it this way?

Response: The study shows an inverse association between alcohol intake and glycated hemoglobin in two independent cohorts, but causality cannot be concluded from observational studies. The finding is not unique. Several observational studies have reported an inverse association between alcohol consumption and HbA1c or insulin resistance (refs 35-37). Moreover, moderate alcohol consumption is shown to reduce HbA1c and insulin resistance in multiple intervention studies as summarized in a meta-analysis (Schrieks IC, diabetes care). Therefore, this finding is consistent with other studies, but the mechanism is currently not resolved. We have now indicated this briefly in the discussion section of the paper as follows; ‘The same is true for alcohol consumption which has been associated with a lower HbA1c level and improved insulin resistance in previous observational and intervention studies, but not particularly with HbA1c independent of glucose [35-37]. The mechanism by which alcohol would improve insulin sensitivity is currently not resolved.’

5. The author further explains the significance of the findings for metabolic or cardiovascular disease will be appreciated. (The significance of the study is unclear because the glycated hemoglobin in the study fluctuates within the normal range) 

Response: The main relevance of the paper is related to the use of HbA1c levels for the diagnosis of diabetes. Since disagreement in diagnosis by HbA1c and glucose is observed in other studies, it is important to understand the contributing factors to HbA1c beyond glucose. Increased HbA1c levels are associated with future CVD incidence. This association is also apparent among individuals with levels of HbA1c in the normal range (Khaw KT (2004) Ann Intern Med 141: 413-420 and Di Angelantonio E (2014) Jama-J Am Med Assoc 311: 1225-1233). The author is right that this point was not particularly articulated in the paper. Therefore we added this sentence in the introduction section: 

“HbA1c has also been shown to be a more accurate marker than measures of glucose for future risk of diabetes complications, such as micro vascular complications and cardiovascular disease (CVD), in individuals with type 2 diabetes [4]. In addition, higher HbA1c is associated with higher risk for CVD and total mortality [5, 6]. This increased CVD and mortality risk with higher HbA1c is already seen at levels in the normal range [5, 6].

6. The conclusion that women's hemoglobin is higher than men's is also worthy of our attention. It would be greatly appreciated if the author could further analyze the pre- and post-menopausal women. (Because previous studies have shown that estrogen treatment can increase glycated hemoglobin levels)

Response: This is an interesting suggestion. The current findings were in contrast with other studies where a higher HbA1c was found in men [REF 15]. And a study were HbA1c was lower in women before menopause as compared to men [REF 31]. However, in previous studies, no adjustment was made for glucose levels. As explained in the discussion section line 263, similar to these previous studies, we also observed a higher HbA1c level in men when we did not correct for glucose. In the current study, the higher HbA1c levels in women were consistently found in both cohorts, in which age, and hence the proportion of postmenopausal women, was different. This suggests that the observed higher HbA1c levels, at similar glucose concentration, were not specific for postmenopausal women or women using estrogen treatment. In addition, in both cohorts only a small proportion of women was premenopausal with 5.5% in the HS and approximately 15% in NHS. We considered that the subsample of premenopausal women was too small to conduct a stratified analysis. 

Reviewer #2: General Comments

This is a very important and interesting study. The replication element of the study greatly increases its impact. It is important to determine the relation between the potential determinants of HbA1c and HbA1c itself.

Specific comments

Methods

A breakdown of Ethnicity of the participants would be very helpful – sorry if I missed this

Response: In the HS cohort, only Caucasians were part of the cohort. In the NHS only 4.5% of individuals were non-Caucasian. We considered that the subsample of non-Caucasians was too small for a subanalysis. 

Results

Please can a Figure be included to display the results of the multivariate regression analysis as described in Table 2. This always aids data appreciation from a reader perspective.

Response: thank you for the suggestion. We are not sure how this would add to the current information since a Figure does not provide specific numbers. Therefore, we prefer the data being presented in a Table.

---

## [Decision Letter · Decision Letter 1]

13 May 2020

Phenotypic and lifestyle determinants of HbA1c in the general population – the Hoorn Study

PONE-D-19-25704R1

Dear MSc Wisgerhof 

We are pleased to inform you that your manuscript has been judged scientifically suitable for publication and will be formally accepted for publication once it complies with all outstanding technical requirements.

With kind regards,

Xianwu Cheng, M.D., Ph.D., FAHA

Academic Editor

PLOS ONE

Additional Editor Comments (optional):

All original concerns have been addressed by the authors.

Reviewers' comments:

Reviewer's Responses to Questions

**Comments to the Author**

1. If the authors have adequately addressed your comments raised in a previous round of review and you feel that this manuscript is now acceptable for publication, you may indicate that here to bypass the “Comments to the Author” section, enter your conflict of interest statement in the “Confidential to Editor” section, and submit your "Accept" recommendation.

Reviewer #1: All comments have been addressed

Reviewer #2: All comments have been addressed

2. Is the manuscript technically sound, and do the data support the conclusions?

Reviewer #1: Yes

Reviewer #2: Yes

3. Has the statistical analysis been performed appropriately and rigorously? 

Reviewer #1: Yes

Reviewer #2: Yes

4. Have the authors made all data underlying the findings in their manuscript fully available?

Reviewer #1: Yes

Reviewer #2: Yes

5. Is the manuscript presented in an intelligible fashion and written in standard English?

Reviewer #1: Yes

Reviewer #2: Yes

6. Review Comments to the Author

Reviewer #1: The authors have shown a lot of effort to improve the manuscript and this should be well appreciated. I found the authors have addressed all my comments carefully by adding more materials in the text. As a result, I now recommend the current form can be accepted for publication without further modification.

Reviewer #2: Please can you accept this excellent paper

I already emailed your office twice to say "ACCEPT"

Kind regards

Dr. Adrian Heald

7. PLOS authors have the option to publish the peer review history of their article (what does this mean?). If published, this will include your full peer review and any attached files.

Reviewer #1: No

Reviewer #2: Yes: Dr. Adrian Heald

---

## [Editor Report · Acceptance letter]

27 May 2020

PONE-D-19-25704R1 

Phenotypic and lifestyle determinants of HbA1c in the general population – the Hoorn Study 

Dear Dr. Wisgerhof:

I am pleased to inform you that your manuscript has been deemed suitable for publication in PLOS ONE. Congratulations! Your manuscript is now with our production department. 

With kind regards,

on behalf of

Associate Prof. Xianwu Cheng 

Academic Editor

PLOS ONE